# Mice Lacking PLAP-1/Asporin Show Alteration of Periodontal Ligament Structures and Acceleration of Bone Loss in Periodontitis

**DOI:** 10.3390/ijms242115989

**Published:** 2023-11-05

**Authors:** Masaki Kinoshita, Satoru Yamada, Junichi Sasaki, Shigeki Suzuki, Tetsuhiro Kajikawa, Tomoaki Iwayama, Chiharu Fujihara, Satoshi Imazato, Shinya Murakami

**Affiliations:** 1Department of Periodontology and Regenerative Dentistry, Osaka University Graduate School of Dentistry, Suita 565-0871, Osaka, Japan; katusando@gmail.com (M.K.); iwayama.tomoaki.dent@osaka-u.ac.jp (T.I.); fujihara.chiharu.dent@osaka-u.ac.jp (C.F.); murakami.shinya.dent@osaka-u.ac.jp (S.M.); 2Department of Periodontology and Endodontolgy, Tohoku University Graduate School of Dentistry, Sendai 980-8575, Miyagi, Japan; shigeki.suzuki.b1@tohoku.ac.jp (S.S.); tetsuhiro.kajikawa.e6@tohoku.ac.jp (T.K.); 3Department of Dental Biomaterials, Osaka University Graduate School of Dentistry, Suita 565-0871, Osaka, Japan; sasaki.junichi.dent@osaka-u.ac.jp (J.S.); imazato.satoshi.dent@osaka-u.ac.jp (S.I.)

**Keywords:** periodontal ligament, knockout animals, periodontitis, extracellular matrix, PLAP-1/Asporin, small leucine-rich repeat proteoglycan

## Abstract

Periodontal ligament-associated protein 1 (PLAP-1), also known as Asporin, is an extracellular matrix protein expressed in the periodontal ligament and plays a crucial role in periodontal tissue homeostasis. Our previous research demonstrated that PLAP-1 may inhibit TLR2/4-mediated inflammatory responses, thereby exerting a protective function against periodontitis. However, the precise roles of PLAP-1 in the periodontal ligament (PDL) and its relationship to periodontitis have not been fully explored. In this study, we employed *PLAP-1* knockout mice to investigate its roles and contributions to PDL tissue and function in a ligature-induced periodontitis model. Mandibular bone samples were collected from 10-week-old male C57BL/6 (WT) and *PLAP-1* knockout (KO) mice. These samples were analyzed through micro-computed tomography (μCT) scanning, hematoxylin and eosin (HE) staining, picrosirius red staining, and fluorescence immunostaining using antibodies targeting extracellular matrix proteins. Additionally, the structure of the PDL collagen fibrils was examined using transmission electron microscopy (TEM). We also conducted tooth extraction and ligature-induced periodontitis models using both wild-type and *PLAP-1* KO mice. *PLAP-1* KO mice did not exhibit any changes in alveolar bone resorption up to the age of 10 weeks, but they did display an enlarged PDL space, as confirmed by μCT and histological analyses. Fluorescence immunostaining revealed increased expression of extracellular matrix proteins, including Col3, BGN, and DCN, in the PDL tissues of *PLAP-1* KO mice. TEM analysis demonstrated an increase in collagen diameter within the PDL of *PLAP-1* KO mice. In line with these findings, the maximum stress required for tooth extraction was significantly lower in *PLAP-1* KO mice in the tooth extraction model compared to WT mice (13.89 N ± 1.34 and 16.51 N ± 1.31, respectively). In the ligature-induced periodontitis model, *PLAP-1* knockout resulted in highly severe alveolar bone resorption, with a higher number of collagen fiber bundle tears and significantly more osteoclasts in the periodontium. Our results demonstrate that mice lacking PLAP-1/Asporin show alteration of periodontal ligament structures and acceleration of bone loss in periodontitis. This underscores the significant role of PLAP-1 in maintaining collagen fibrils in the PDL and suggests the potential of PLAP-1 as a therapeutic target for periodontal diseases.

## 1. Introduction

Periodontal ligament-associated protein 1 (PLAP-1)/Asporin is an extracellular matrix protein (ECM) that is preferentially expressed in periodontal ligaments (PDL) [1]. PLAP-1 belongs to the small leucine-rich repeat proteoglycan (SLRP) family class I with decorin and biglycan. It has a consensus motif containing four cysteine residues in the N-terminal region, two cysteine residues in the C-terminal region, and a central region harboring ten leucine-rich repeat (LRR) domains [2,3]. PLAP-1 inhibits the cytodifferentiation and mineralization of PDL cells by inhibiting BMP-2 [3] and that of chondrocytes by inhibiting TGF-β [4]. Therefore, PLAP-1 plays crucial roles in the maintenance of periodontal tissue homeostasis.

The PDL is a unique tissue that is surrounded by two hard tissues, teeth and alveolar bone; it maintains the properties of soft tissue as a ligament while receiving mechanical stress such as occlusal force. The PDL is composed of various cell components including PDL fibroblasts and various ECM proteins such as collagen [5]. The ECM in the PDL not only functions as a component that fills the space between cells, but it also exhibits various biological activities, with involvement in cell proliferation, differentiation and fibrosis [6]. The ECM is directly involved in the formation and maturation of collagen type I, which is the most common fiber component in the PDL [7]. Biglycan and decorin, which are also members of SLRP class I, are involved in the maintenance of cross-link homeostasis by interacting with collagen through core proteins and in fibrogenesis [8,9,10]. PLAP-1 binds to collagen type I, suggesting that PLAP-1 may be directly involved in functions such as fibrogenesis in the PDL and collagen strength [11].

Periodontitis is a chronic inflammatory disease that is associated with bacterial infection and mechanical stress [12,13]. It can result in the destruction of the PDL and periodontal tissue. We previously reported that PLAP-1 may have a defensive role in periodontitis lesions by suppressing pathophysiologic TLR signaling [14]. However, the precise function of PLAP-1, which plays an important role in maintaining PDL homeostasis, in the progression of periodontitis is unknown.

To better understand the role of PLAP-1 in the PDL, in this study, we investigated changes in the periodontal tissue of *PLAP-1* knockout (*PLAP-1* KO) mice [15]. To elucidate the role of PLAP-1 in the function of collagen fiber bundles that make up the PDL, we used techniques to observe the strength of the PDL and the reaction associated with the progression of periodontitis. We analyzed the role and function of PLAP-1 in the morphology and strength of PDL tissues.

## 2. Results

### 2.1. Increased PDL Space in PLAP-1 KO Mice

The role of PLAP-1 in periodontal tissues in vivo was investigated using *PLAP-1* KO mice [15]. We performed μCT imaging of the maxillary alveolar bone from 10-week-old male WT and *PLAP-1* KO mice. Alveolar bone level was measured as the sum of the distances from the cement–enamel junction to the alveolar apex for the first molar centrifugal root, second molar proximal root, second molar centrifugal root, and third molar root (Figure 1A, yellow lines). No significant differences were observed in alveolar bone level in *PLAP-1* KO mice compared with WT mice (Figure 1B). The morphology of maxillary second molars of 12-week-old WT and *PLAP-1* KO mice was observed using μCT imaging. The angle of the root axis of the proximal and centrifugal buccal roots was measured as root detachment (Figure 1C, yellow lines); there was no significant difference in the maxillary second molar root detachment of *PLAP-1* KO mice compared with WT mice (Figure 1D). Next, the second molar PDL space was observed by reconstructing the second molar periodontal space in the maxillary bone of 12-week-old WT and *PLAP-1* KO mice from μCT (Figure 1E). The results showed a significant increase in the volume of the maxillary second molar PDL space in *PLAP-1* KO mice compared with WT mice (Figure 1F). In addition, a vertical section through the center of the buccal side root was prepared from the μCT scan of the second molar in 12-week-old WT and *PLAP-1* KO mice (Figure 1G). Measurement of the PDL area in the vertical section revealed a significant increase in PDL space of *PLAP-1* KO mice compared to WT mice (Figure 1H). Hematoxylin and eosin (HE) staining of the periodontium in *PLAP-1* KO mice also confirmed an enlargement of the PDL space in comparison to WT mice (Figure 1I).

### 2.2. Altered Collagen Bundles and Extracellular Matrix Expression in the PDL of PLAP-1 KO Mice

To observe the collagen fiber bundles comprising the PDL, thin sections of maxillary periodontal tissue from 10-week-old male WT and *PLAP-1* KO mice were prepared and subjected to picrosirius red staining (Figure 2A). The density of the collagen fiber bundles in the PDL of *PLAP-1* KO mice was observed to be reduced compared with that of WT mice. We assessed the sparseness of periodontal fiber bundles in the proximal region of the first molar, excluding vessels with a diameter of 20 µm or more (Figure 2A, yellow region). The results showed a significant decrease in the density of collagen fiber bundles in the PDL of *PLAP-1 KO* mice compared with WT mice (Figure 2A).

We then analyzed the expression of ECM-related genes in the PDL tissues of molars of 10–13-week-old WT and *PLAP-1* KO mice (Figure 2B). We first confirmed knock-out of PLAP-1 gene expression in *PLAP-1* KO mouse PDL tissues (Figure 2B). There was no difference in the gene expression of *Col1a1*, *Col3a1*, *BGN,* and *DCN* in the PDL of *PLAP-1* KO mice compared with WT mice (Figure 2B).

We next analyzed ECM-related protein expression in the PDL. Sections of maxillary periodontal tissue from 10-week-old male WT and *PLAP-1* KO mice were prepared and fluorescence immunostaining was performed with anti-PLAP-1, anti-Col1, anti-Col3, anti-BGN, and anti-DCN antibodies (Figure 2C). Quantification of fluorescence intensity in the PDL revealed a significant increase in Col3, BGN, and DCN protein expression in the PDL of *PLAP-1* KO mice compared with WT mice (Figure 2C).

### 2.3. Altered Structure and Mechanical Properties of Collagen Fibrils of PDL in PLAP-1 KO Mice

To examine the morphology of collagen fibrils in more detail, periodontal tissues of 6- and 12-week-old male WT and *PLAP-1* KO mice were observed by transmission electron microscopy (TEM). Cross-sectional images of the collagen fibrils of the PDL are shown in Figure 3A. We measured the diameter of the collagen fibrils of the PDL and performed quantitative analysis. The results showed that the diameter of collagen fibrils in *PLAP-1* KO mice was significantly increased at 6 weeks and 12 weeks of age compared with that of WT mice (Figure 3B,C).

To assess the strength of the PDL, a model was created in which the PDL was physically torn off for tooth extraction as described by Trombetta-eSilva et al. (Figure 4A) [16]. Mandibular bone samples from 12-week-old WT and *PLAP-1* KO mice were fixed, and a wire was inserted through the bifurcation of the first molar root. The maximum stress required for tooth extraction was subsequently determined. The maximum stress during tooth extraction via PDL traction in WT and *PLAP-1* KO mice was 16.51 N ± 1.31 and 13.89 N ± 1.34, respectively (Figure 4B). These results revealed a significant reduction in the maximum stress needed for tooth extraction by PDL traction in *PLAP-1* KO mice compared to WT mice.

### 2.4. Increased Alveolar Bone Resorption in PLAP-1 KO Mice in Ligature-Induced Periodontitis

To assess the response of the altered PDL structure in *PLAP-1* KO mice in periodontitis, we analyzed alveolar bone resorption in a ligature-induced periodontitis model in *PLAP-1* KO mice. After 14 days of silk ligature threading, the maxillary bone was retrieved and μCT imaging and sections were prepared. Images of bone resorption induced by the ligature in WT and *PLAP-1* KO mice were analyzed. Severe alveolar bone resorption was observed in P*LAP-1* KO mice compared with WT mice in the ligature groups (Figure 5A). Alveolar bone resorption was measured from the cement–enamel junction to the alveolar apex distance as the sum of the first molar centrifugal root, second molar proximal root, second molar centrifugal root, and third molar root measurements (Figure 5A, yellow lines). The results showed that significant bone resorption occurred in WT and *PLAP-1* KO mice on the ligatured side, while there was no difference in alveolar bone resorption on the control side (Figure 5B). Furthermore, there was significantly greater bone resorption in *PLAP-1* KO mice compared with WT mice in the ligature groups.

We prepared thin sections of periodontal tissue and observed collagen fiber bundles on the ligature side by picrosirius red staining. Many collagen fiber bundle tears were observed in *PLAP-1* KO mice (Figure 5C, arrows). To examine osteoclasts, which are responsible for alveolar bone resorption, we prepared thin sections of periodontal tissue and performed TRAP staining. Osteoclasts were observed on the ligature side in both *PLAP-1* KO and WT mice (Figure 5D). We quantified the numbers of osteoclasts around the proximal root of the second molar tooth, and the results showed a significantly increased number of osteoclasts in *PLAP-1* KO mice compared with WT mice in the ligature groups (Figure 5E).

## 3. Discussion

In this study, we demonstrated the functional importance of PLAP-1, which is specifically expressed in the PDL, in periodontal fibrogenesis, and the functional maintenance of periodontal tissue homeostasis.

PLAP-1 is a SLRP class I protein that was identified in our laboratory and is specifically expressed in the PDL. PLAP-1 is expressed in the dental follicle, which differentiates into cementum, PDL, and intrinsic alveolar bone during tooth development [3,17]. To investigate the effect of *PLAP-1* KO on the root and alveolar bone in mice, μCT and HE staining of the periodontal tissues were performed. We found no obvious abnormality in alveolar bone resorption and root morphology in *PLAP-1* KO mice.

To quantitatively examine any abnormalities of collagen fibrils, the collagen fibrils in the PDL of *PLAP-1* KO mice were observed by TEM, which allows cross-sectional observation. The results showed that the diameter of collagen fibrils, which constitute the PDL, was increased in *PLAP-1* KO mice. PLAP-1 has a collagen-binding site at LRR10 [11]. PLAP-1 directly binds to collagen and plays an important role in the maturation and cross-linking of collagen fibrils. In *PLAP-1* KO mice, abnormal meshwork structure and maturation caused collagen fibrils to tear and increase in diameter.

Previous studies reported that a large proportion of collagen fibrils with small diameters have a greater ability to withstand plastic deformation [18] and that the tensile strength of skin is reduced in *decorin* KO mice despite the large diameter of collagen fibrils in the skin [19,20,21]. In this study, we performed traction on the PDL of *PLAP-1* KO mice and evaluated the strength of the PDL using the method of Trombetta-e Silva et al. [16]. The results showed that the mechanical strength of the PDL was reduced in *PLAP-1* KO mice. Thus, PLAP-1 has significant effects on the maintenance of collagen fibrils in the normal PDL and on the strength of the PDL. Picrosirius red staining of collagen fibrils revealed a decrease in the density of collagen fibrils in the PDL of *PLAP-1* KO mice. Fluorescence immunostaining showed that the expression level of type I collagen, which accounts for most of the fibrous collagen in the PDL, did not change, while the expression level of type III collagen increased. These findings suggest an altered localization of collagen fibril bundles in *PLAP-1* KO mice. Some studies reported that PLAP-1 suppresses chondrogenesis in osteoarthritis through TGF-β signaling [22,23]. Abnormalities in tissue fibrosis are regulated through TGF-β signaling [24,25], suggesting that TGF-β signaling may also be involved in the abnormalities of collagen fibers in the PDL of *PLAP-1* KO mice. PLAP-1 is an inhibitory regulator of TGF-β [4], and thus it may be involved in collagen formation and other processes through its downstream Smad signaling. It is possible that PLAP-1 negatively regulates TGF-β in vivo. This strongly suggests that Smad signaling is also dysregulated in vivo. Future studies should explore the expression changes in TGF-β-related genes and proteins in the PDL of *PLAP-1* KO mice.

The PDL of *PLAP-1* KO mice showed decreased strength and an enlarged PDL cavity, suggesting that occlusal trauma may have become more severe because of the increased alveolar bone resorption and increased number of osteoclasts. PLAP-1 negatively regulates TRL2- and TRL4-mediated inflammatory responses in PDL cells [14]; *PLAP-1* KO mice may lack this inhibitory mechanism, leading to the severity of periodontitis observed in this study. Further studies are required to investigate whether *PLAP-1* KO mice develop periodontitis using occlusal trauma models and old age mouse models.

In periodontal tissues, biglycan is highly expressed in dentinoblasts [26,27] and decorin is widely detected in periodontal tissues and also in the PDL [28,29]. No major abnormalities in periodontal tissues were observed in *biglycan* and *decorin* KO mice [30,31,32], but abnormalities in the network structure of collagen protofibrils in the PDL and an increase in the diameter of collagen protofibrils were observed in *decorin* KO mice [30,31,33]. Additionally, *biglycan* KO mice show an increase in the diameter of collagen fibrils in the PDL [19,33]. In this study, *PLAP-1* KO mice showed an increased diameter of collagen protofibrils and an abnormal reticular structure, suggesting that SLRP class 1 function for collagen protofibrils may be partially duplicated. However, in the PDL, PLAP-1 is highly expressed compared with biglycan and decorin, suggesting that PLAP-1 plays a major role in the PDL and in collagen formation [34]. In the PDL of *PLAP-1* KO mice, biglycan and decorin expressions were increased at the protein level. However, the slowed protein metabolism partially complements PLAP-1 function but does not compensate for it quantitatively or functionally. Therefore, we believe that various collagen abnormalities and reductions in strength may have occurred in *PLAP-1* KO mice.

A previous study reported a decrease in the diameter of collagen protofibrils and an increase in the strength of the skin of *PLAP-1* KO mice [35]. While this phenotype differs from that observed in the PDL of *PLAP-1* KO mice, the expression of PLAP-1 is extremely high in the PDL compared with skin, and thus the effects of *PLAP-1* KO may have been more profound in or specific to the PDL. Additionally, the PDL is a special tissue that is constantly subjected to loads such as occlusal forces, and the role of PLAP-1 is thought to be important in maintaining strength in the PDL to withstand these loads.

Further analysis of the role of PLAP-1 in collagen formation and in the homeostasis of periodontal tissues will help to clarify the value of PLAP-1 as a potential therapeutic agent for periodontal disease.

## 4. Materials and Methods

### 4.1. Animals 

All animal experiments were approved by the Institutional Animal Care and Use Committee of Osaka University Graduate School of Dentistry (permit number 30-010-0). C57BL/6 WT mice (10–12 weeks old) were purchased from Japan SLC Inc. (Shizuoka, Japan). We generated *Plap-1*^−/−^ (*PLAP-1* KO) mice using homologous recombination in embryonic stem cells following standard procedures [36]. The genomic locus encoding the murine *PLAP-1* gene was targeted by a replacement DNA construct, leading to disruptions in exon 2 and 3 caused by the neomycin resistance cassette. Two independent ES cell clones were utilized to produce chimeras capable of transmitting the targeted allele to the germline when mated with C57BL/6 females [15]. The number of animals used in each experiment is shown in Appendix A. 

### 4.2. Quantification of Alveolar Bone Loss

WT and *PLAP-1* KO mice were refluxed and fixed using 4% paraformaldehyde phosphate (4% PFA) under pentobarbital sodium anesthesia, and the maxilla was collected. The maxilla was subjected to tomography using 3D micro-X-ray CT R_mCT2 (RIGAKU, Tokyo, Japan) for laboratory animals, and the obtained 3D images were converted to 2D using the 3D image analysis software TRI/3D-BON; http://www.ratoc.com/, accessed on 17 September 2023 (Ratoc System Engineering, Tokyo, Japan). The sum of the distance from the cement–enamel junction to the alveolar bone crest of the maxillary first molar distal root, second molar mesial and distal root, and the third molar was analyzed in the 2D images using the image analysis software WinROOF; https://www.mitani-visual.jp/en/ (accessed on 17 September 2023, Mitani, Fukui, Japan).

### 4.3. Quantification of Root Angle and PDL Volume

Twelve-week-old WT and *PLAP-1* KO mice were refluxed and fixed in 4% PFA under pentobarbital sodium anesthesia, and the maxilla and mandible were collected. Tomography was performed with a high-resolution 3D X-ray microscope, SKYSCAN1272 (BRUKER Japan, Kanagawa, Japan), and the obtained 3D image of the maxillary second molar was converted into a 2D image using analysis software; CTvox; https://www.bruker.com/, accessed on 17 September 2023 (BRUKER Japan). Quantitative analysis was performed by measuring the angle of the root axis between the mesial buccal root and the distal buccal root. The volume of the PDL was analyzed quantitatively from the obtained 3D images using analysis software; CTAn; https://www.bruker.com/ (BRUKER Japan). The area of PDL was quantitatively analyzed in the section through the center of the buccal side root of the second molar using the image analysis software Win-ROOF; https://www.mitani-visual.jp/en/.

### 4.4. Histological Analysis and Immunohistochemistry

Ten-week-old WT and *PLAP-1* KO mice were refluxed and fixed in 4% PFA under pentobarbital sodium anesthesia, and the maxilla was collected. After decalcification with EDTA, the tissue was embedded in paraffin wax. Serial 8 µm-thick sections were cut in the transverse direction for the first molars and mounted onto slides.

Representative sections from each block were stained with hematoxylin and eosin; sections were also stained using the Picrosirius Red Stain Kit (0.1% of Sirius red in saturated aqueous picric acid) (Polysciences, Philadelphia, PA, USA) and the TRAP/ALP Stain Kit (WAKO, Osaka, Japan). To assess the density of collagen fiber bundles in the PDL, the area from the alveolar apex to the root apex of the PDL, 300 µm from the alveolar apex, excluding vessels larger than 20 µm in diameter, was binarized and analyzed quantitatively. To quantitatively analyze the number of osteoclasts, we counted the number of osteoclasts on the alveolar bone surface around the first molar centrifugal root of the first molar in a 1000 µm square.

Paraffin-embedded sections were rinsed for 5 min in distilled water. Sections were incubated with chondroitinase ABC from Proteus vulgaris (Sigma-Aldrich Co., St. Louis, MO, USA) for 2 h at room temperature (the same treatment was performed for staining with decorin and biglycan antibodies). Thin sections were reacted with NxGen (BIOCARE Medical LLC., Pacheco, CA, USA) in Target Retrieval Solution (Agilent Technologies, Santa Clara, CA, USA) at 90 °C for 3 min. The sections were also blocked in Dulbecco phosphate buffer (Wako) containing 3% bovine serum albumin (Sigma-Aldrich) for 1 h at room temperature. Primary antibodies, including goat anti-mouse PLAP-1 antibody (1:100, Abcam plc, Cambridge, Cambridgeshire, UK), rabbit anti-mouse type I collagen antibody (1:100, Abcam plc), rabbit anti-mouse type III collagen antibody (1:100, Abcam plc), rabbit anti-mouse decorin antibody (LF-113, 1:2000, kind gifts from Dr. Larry Fisher, NIDCR/NIH, Bethesda, MD, USA), and rabbit anti-mouse biglycan antibody (LF-159,1:2000, kind gifts from Dr. Larry Fisher), were reacted at 4 °C for 18 h, and the samples were then washed with 0.05% polyoxyethylene (20) sorbitan monolaurate (Wako) containing PBS. Secondary antibodies, including Alexa Fluor Plus 488-conjugated donkey anti-goat IgG antibody (1:200, Abcam), Alexa Fluor Plus 488-conjugated donkey anti-rabbit IgG antibody (1:200, Abcam), and Alexa Fluor Plus 594-labeled donkey anti-rabbit IgG antibody (1:200 (Abcam), were reacted for 30 min in a dark room at room temperature. Nuclear staining was performed using DAPI (Sigma-Aldrich) for 10 min at room temperature. Both primary and secondary antibodies were diluted in 3% bovine serum albumin in PBS. The samples were sealed using Prolong™ Diamond Antifade Mountant (Thermo Fisher Scientific Inc., Waltham, MA, USA) and then analyzed by a confocal microscope SP8 LIGHTNING (Leica Microsystems, Wetzlar, Germany). For quantitative analysis of fluorescent immunostaining images, three different representative sites were extracted from the PDL and binarized. Negative controls were also binarized using the same technique, and the expression levels were subtracted and the average value of the three sites per unit area was measured. Each sample was measured repeatedly in three different sections of an individual by an examiner named MK. The average value was then calculated to represent the intensity of expression.

### 4.5. Gene Expression Analysis

The excised maxillary tissue was immersed in 3 mL of RNAlater Stabilization Solution (Thermo Fisher Scientific). After the soft tissue, including the gingiva, was detached from the maxillary tissue, the maxillary first molar was extracted with a 26 G injection needle (Terumo, Tokyo, Japan). The maxillary first molar was grasped with tweezers and the PDL tissue around the proximal root of the first molar was scraped and collected using a micro sharp spoon (Fine Science Tools, Vancouver, Canada). Total RNA was extracted from the collected tissues using the RNeasy Micro Kit (QIAGEN, Hilden, Germany). Total RNA was extracted from cultured cells using the PureLink RNA Mini Kit (Thermo Fisher Scientific). Reverse transcription reactions were performed using the High-Capacity RNA-to-cDNA Kit (Thermo Fisher Scientific) to prepare complementary stranded DNA (cDNA). The obtained cDNAs were used as templates for real-time PCR analysis using the gene-specific primers listed in Table 1. PCR reactions were performed using Fast SYBR Green Master Mix (Thermo Fisher Scientific) with the StepOnePlus Real-Time PCR System (Life Technologies, Carlsbad, CA, USA). The expression of the hypoxanthine phosphoribosyltransferase (Hprt) gene was used as an internal control.

### 4.6. Transmission Electron Microscopic (TEM) Analysis

After reflux fixation with 2% paraformaldehyde phosphate buffer (2% PFA), the maxillary periodontal tissue was excised and immersed in 2% PFA overnight. The maxillary periodontal tissue was immersed in 2% glutaraldehyde phosphate buffer containing 0.1 M phosphate buffer and then immersed in a 2% osmium solution containing 0.1 M phosphate buffer at 4 °C for 3 h. After dehydration, the maxillary periodontal tissue was infiltrated with propylene oxide twice for 30 min, immersed in a mixture of resin (Nisshin, Tokyo, Japan) and propylene oxide for 6 h, and further immersed overnight in a resin. After polymerization at 60 °C for 48 h, ultra-thin sections with a thickness of 70 nm were prepared using an ultramicrotome (Leica Microsystems). The ultrathin sections were observed and photographed with a JEM-1400Plus transmission electron microscope (JEOL Ltd., Akishima, Japan). A total of 800 collagen primordial fiber diameters at eight randomly selected sites were measured and quantified using the image analysis software WinROOF; https://www.mitani-visual.jp/en/. 

### 4.7. Periodontal Traction Model

Twelve-week-old WT and *PLAP-1* KO mice were anesthetized with sodium pentobarbital, and the mandible was excised and immersed in PBS. A 0.2 mm diameter tungsten wire (Niraco, Tokyo, Japan) was threaded through the first molar root bifurcation as described by Trobetta-e Silva et al. [16]. A small tabletop tester, SHIMADZU EZ-S 500N (Shimadzu, Kyoto, Japan), was used to fix the CR around the mandible, and the tungsten wire was towed at a crosshead speed of 1 mm/min. The maximum stress applied before the first molar was extracted was measured.

### 4.8. Ligature-Induced Periodontitis

Eight-week-old WT mice and *PLAP-1* KO mice were anesthetized by intraperitoneal injection of sodium pentobarbital anesthesia; all efforts were made to minimize suffering. An 8-0 silk ligature (Akiyama Seisakusyo, Tokyo, Japan) was tied around the maxillary left second molar as described by Abe et al. [37] on day 0. After 14 days, mice were euthanized, and the maxillae were removed for micro-computed tomography and histological analysis. Using the image analysis software WinROOF; https://www.mitani-visual.jp/en/, the distance of the root apex from the cement–enamel junction to the alveolar apex in the two-dimensional images was measured, and the sum of the measurements at the first molar centrifugal root, the second molar pericentric and centrifugal roots, and the third molar root was the alveolar bone resorption.

### 4.9. Statistical Analysis

Statcel version 4 software (OMS Publishing, Saitama, Japan) was used for statistical analysis. Data are shown as the mean ± SD. Statistical analyses were performed using the Student’s *t*-test for paired comparisons and one-way analysis of variance for multiple comparisons with Bonferroni for the posttest. *P* < 0.05 was considered statistically significant.

## Figures and Tables

**Figure 1 ijms-24-15989-f001:**
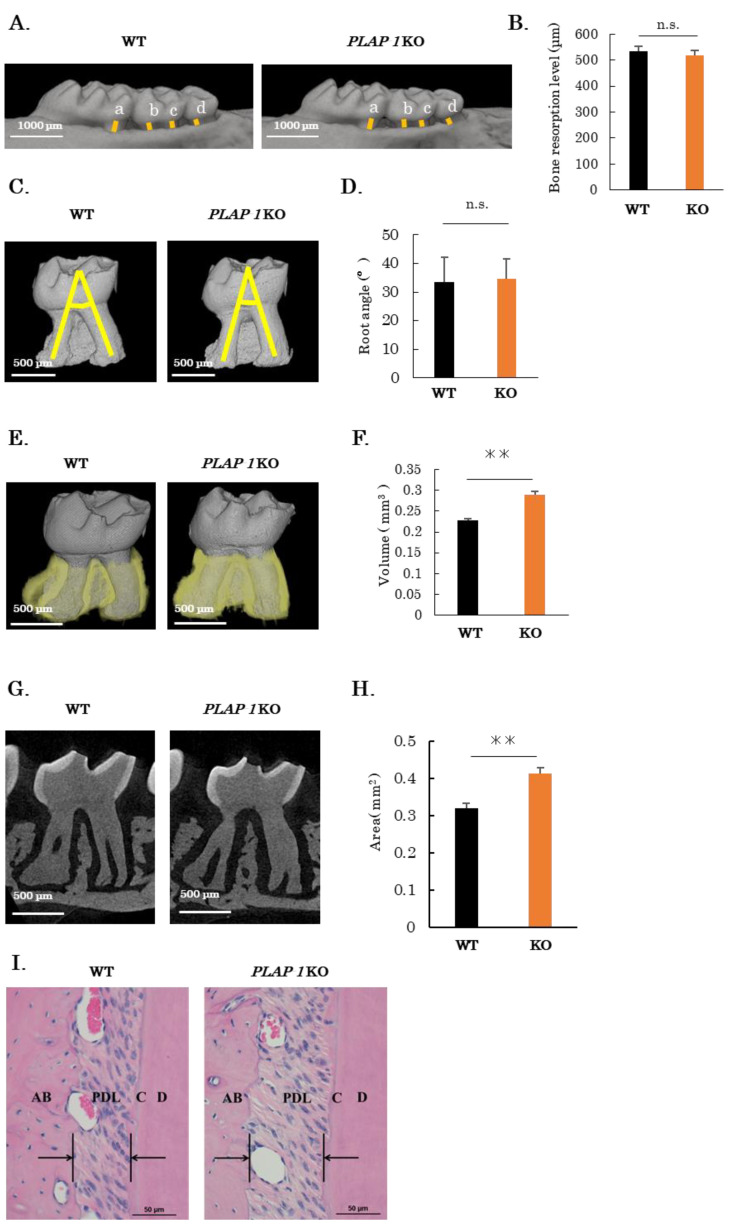
Increased periodontal ligament space in *PLAP-1* KO mice. (**A**) µCT imaging of maxillary alveolar bone collected from 10-week-old male WT and *PLAP-1* KO mice (*n* = 4 in each group). (**B**) The sum of the measurements (a + b + c + d) at the distance from the cement–enamel junction to the alveolar apex at the first molar centric root (a), second molar proximal root (b), second molar centric root (c), and third molar root (d) was determined as the alveolar bone resorption level. (**C**) High-resolution μCT imaging of maxillary second molar of 12-week-old male WT and *PLAP-1* KO mice. (**D**) The angle of the root axis of the proximal and distal buccal roots was measured as the degree of root detachment (*n* = 3 in each group). (**E**,**F**) The volume of the maxillary second molar periodontal ligament space in 12-week-old male WT and *PLAP-1* KO mice (*n* = 4 in each group). (**G**) Representative images of cross-sections obtained through the central portion of the buccal side root in high-resolution μCT images of the second molar in 12-week-old male WT and *PLAP-1* KO mice. (**H**) The area of the periodontal ligament space in the maxillary second molar of 12-week-old male WT and *PLAP-1* KO mice was calculated (*n* = 3 in each group). (**I**) Representative images of hematoxylin and eosin staining of the periodontium from 8-week-old male WT and *PLAP-1* KO mice. Scale bar: 50 μm. Results shows the mean ± SD. **: *p*  <  0.01 and n.s. = not significant.

**Figure 2 ijms-24-15989-f002:**
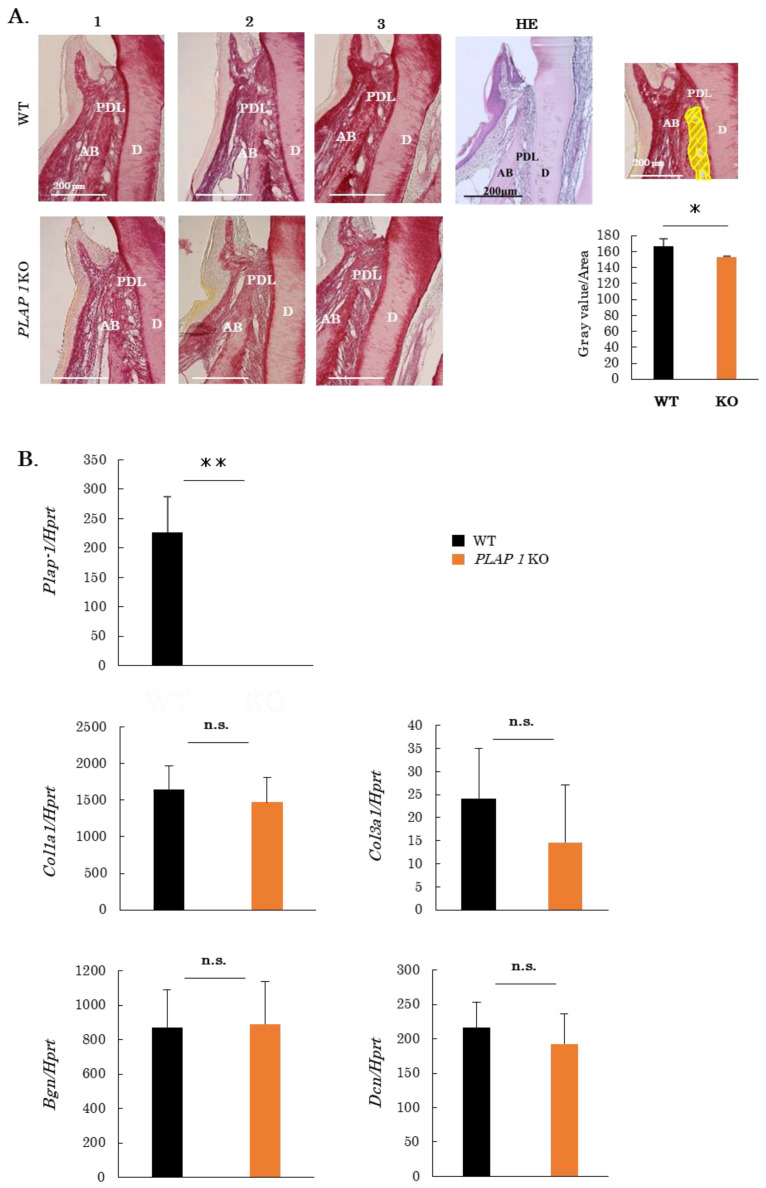
Altered collagen bundles and expression of extracellular matrix components in the PDL of *PLAP-1* KO mice. (**A**) Picrosirius red-stained images of periodontal tissue from 10-week-old WT and *PLAP-1* KO mice, and HE staining of a WT are shown. AB: alveolar bone, D: dentin, and PDL: periodontal ligament. Quantitative analysis of the sparseness and density of periodontal ligament fiber bundles is shown in the yellow highlighted region (vessels greater than 20 µm in diameter were excluded) (*n* = 4 in each group). Data are shown as the mean ± SD in triplicate assays. *: *p* < 0.05. (**B**) Total RNA was extracted from the maxillary first molar periodontal ligament of 10–13-week-old male WT and *PLAP-1* KO mice, and gene expression was analyzed by real-time PCR (*n* = 4 in each group). Data represent the mean ± SD from triplicate assays. Col1a1: collagen type 1 alpha1, Col3a1: collagen type 3 alpha1, Bgn: biglycan, and Dcn: decorin. **: *p*  <  0.01 and n.s. = not significant. (**C**) Fluorescence immunostaining images of ECM components in the periodontal tissue of the maxilla of 10-week-old male WT and *PLAP-1* KO mice. AB: alveolar bone, D: dentin, and PDL: periodontal ligament. Quantification of fluorescence intensity in the periodontal ligament is shown in the right panel (*n* = 3 in each group). Data represent the mean ± SD from triplicate assays. Col1: collagen type 1, Col3: collagen type 3, BGN: biglycan, and DCN: decorin. *: *p* < 0.05 and **: *p* < 0.01.

**Figure 3 ijms-24-15989-f003:**
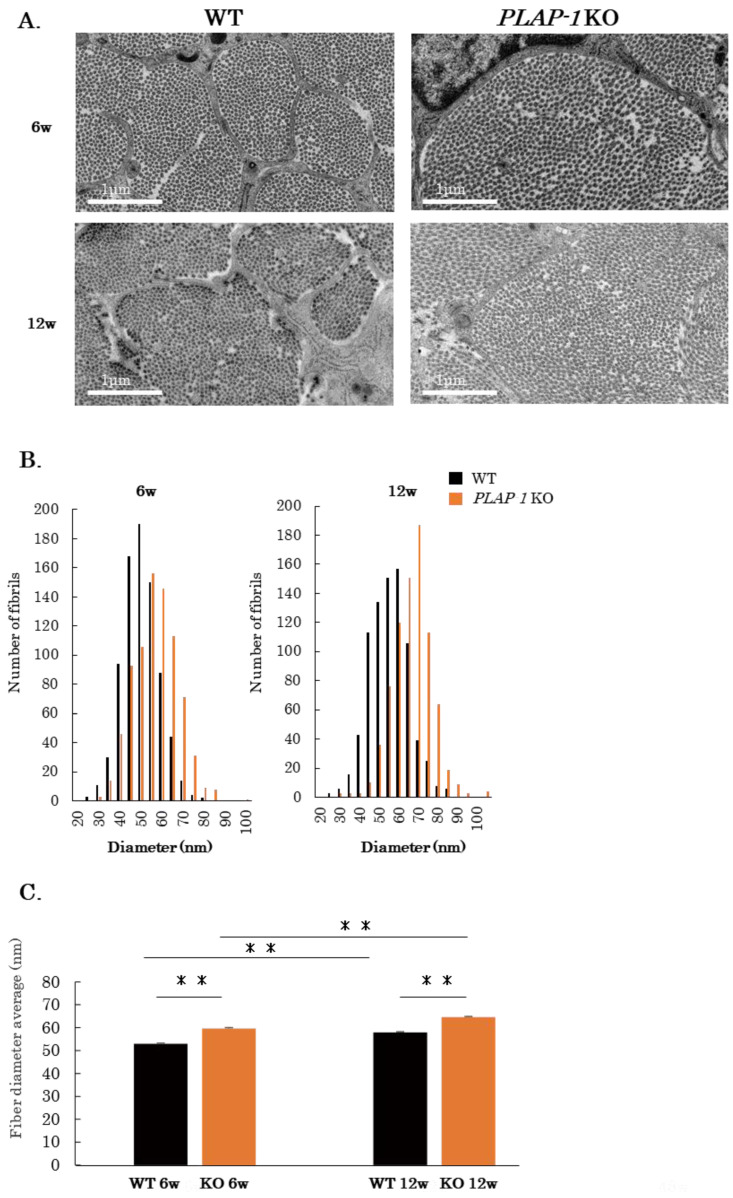
Altered structure of the PDL of *PLAP-1* KO mice. (**A**) Transmission electron microscope (TEM) images (×19,000) of the maxillary periodontal ligament of 6- and 12-week-old WT and *PLAP-1* KO mice. Scale bar: 1 μm. (**B**,**C**) The diameter of collagen fibrils at randomly selected locations was measured at 6- and 12-week-old WT and *PLAP-1* KO mice (*n* = 4 in each group, 200 fibrils were counted in each sample) and quantitatively analyzed. Values represent the mean ± SD from triplicate assays. **: *p* < 0.01.

**Figure 4 ijms-24-15989-f004:**
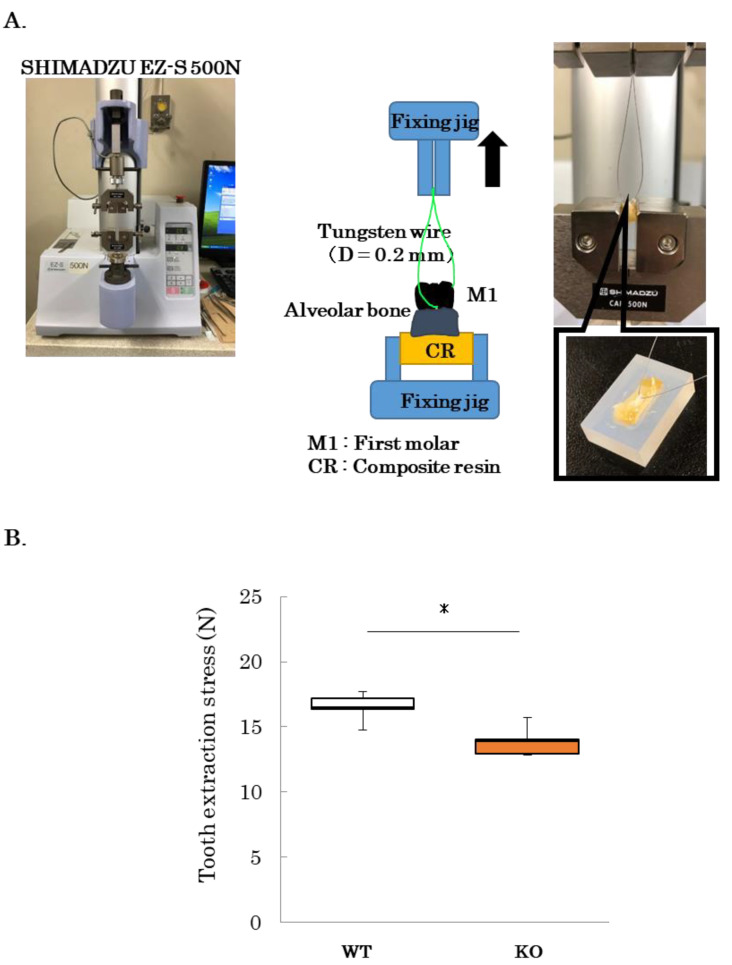
Periodontal ligament traction test with tooth extraction in *PLAP-1* KO mice. (**A**) Image of periodontal ligament traction test. (**B**) The maximum stress required to extract a tooth is shown by mandibular first molar traction in 12-week-old WT and *PLAP-1* KO mice (*n* = 4 in each group). Data represent the mean ± SD in triplicate assays. *: *p* < 0.05.

**Figure 5 ijms-24-15989-f005:**
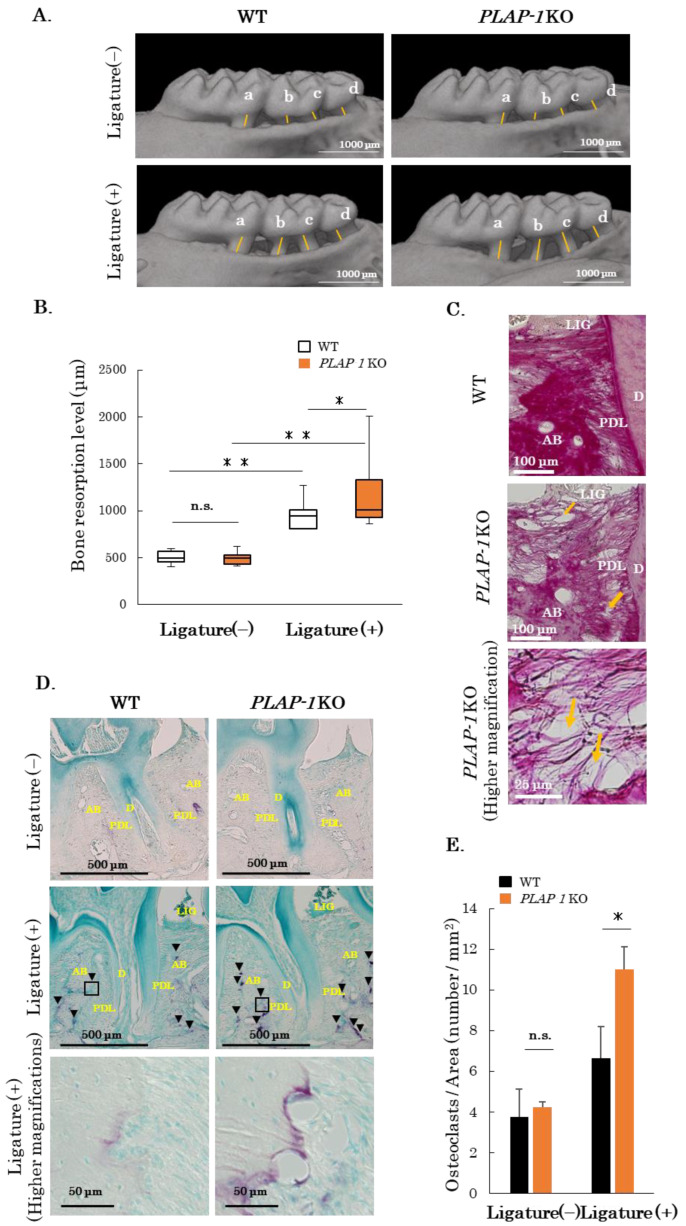
Increased alveolar bone resorption in *PLAP-1* KO mice with ligature-induced periodontitis. (**A**) μCT images of maxillary periodontal tissue of WT and *PLAP-1* KO mice on the ligated and non-ligated sides are shown. (**B**) Distance from cement–enamel junction to the alveolar bone crest was measured at four points indicated on the first molar to the third molar (a + b + c + d) (WT: *n* = 11, *PLAP-1* KO: *n* = 12). (**C**) Picrosirius red-stained images of periodontal tissue from WT and *PLAP-1* KO mice on the ligation side are shown (arrows: collagen fiber bundle rupture). (**D**) TRAP-staining of periodontal tissue from WT and *PLAP-1* KO mice on the ligated and non-ligated side are shown (arrowheads: osteoclasts). (**E**) The number of osteoclasts on the alveolar bone surface around the distal root of the first molar was measured (*n* = 6 in each group). Data represent the mean ± SD in triplicate assays. AB: alveolar bone, D: dentin, PDL: periodontal ligament, and LIG: ligature. *: *p* < 0.05, **: *p* < 0.01, and n.s. = not significant.

**Table 1 ijms-24-15989-t001:** Nucleotide sequences of primers used for polymerase chain reaction (PCR) analysis.

GenBank Acc.	Gene	Sequence
NM_025711	*PLAP-1*	5’-ATGATGACGATAACGATGATGACGA-3’ 5’-TGTTGTTTGGAACCGATGTCAGA-3’
NM_007742	*Col1a1*	5’-CAGGGTATTGCTGGACAACGTG-3’ 5’-GGACCTTGTTTGCCAGGTTCA-3’
NM_009930	*Col3a1*	5’-CAGGCCAGTGGCAATGTAAAGA-3’ 5’-CTCATTGCCTTGCGTGTTTGATA-3’
NM_007542	*Bgn*	5’-GATGATTGAGAATGGGAGCCTGA-3’ 5’-TCCGAAGCCCATAGGACAGAAG-3’
NM_001190451	*Den*	5’-CTGGGCTGGCACAGCATAAGTA-3’ 5’-CGGACAGGGTTGCCGTAAAG-3’
NM_013556	*Hprt*	5’-TTGTTGTTGGATATGCCCTTGACTA-3’ 5’-AGGCAGATGGCCACAGGACTA-3’

## Data Availability

The data presented in this study are available on request from the corresponding author.

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
