# Peer review of "Mice Lacking PLAP-1/Asporin Show Alteration of Periodontal Ligament Structures and Acceleration of Bone Loss in Periodontitis"

_ijms, 2023, doi:10.3390/ijms242115989_

Round 1

Reviewer 1 Report

Comments and Suggestions for Authors

The authors present an animal study on the effect of PLSAP-1/Asporin on the status periodontal tissues. The topic is of high relevance in the field. However, major revisions are required before this manuscript can be considered for publication, mainly in relation to the materials and methods:

1. A flow chart is needed to show the number of animals in each experimental part of the study. The manuscript as it is lacks information on the number of animals, and this needs to be clarified. 

2. What was the volume of RNA later? How did the authors test the quality and quantity of RNA? How was the reference gene chosen? The table on PCR primers is of low resolution and needs to be redone. 

3. generation of the knockout mice needs to be described, even if briefly. To just refer to other reference on the procedure is not sufficient. 

4. What software was used in statistics?

5. Description of the inter-/ intra-examiner calibration needs to be addressed regarding the qualitative histological analysis. Furthermore, initials of the examiner needs to be written in this section. Was the examiner blinded?

6. The abstract needs to be divided into: background, materials and methods, results and conclusions. The authors need to include the statistical values as well in the abstract. 

Comments on the Quality of English Language

Minor English language edits are needed, in terms of grammar. 

Author Response

Reviewer #1

The authors present an animal study on the effect of PLAP-1/Asporin on the status periodontal tissues. The topic is of high relevance in the field. However, major revisions are required before this manuscript can be considered for publication, mainly in relation to the materials and methods:

1.A flow chart is needed to show the number of animals in each experimental part of the study. The manuscript as it is lacks information on the number of animals, and this needs to be clarified.

Response: In alignment with the reviewer's comments, we have added a Supplemental Table 1 displaying the number of animals in each experiment. Furthermore, we have diligently reviewed the figure legends to ensure accurate descriptions of the number of animals.

  1. What was the volume of RNA later? How did the authors test the quality and quantity of RNA? How was the reference gene chosen? The table on PCR primers is of low resolution and needs to be redone.

              Response: Thank you for bringing this to our attention. The volume of RNAlater Stabilization Solution was 3 ml, and we have now included this volume in the Materials and Methods section.

We consistently assess the quantity of RNA extracted from tissues by measuring OD260/OD280 and OD260/OD230 values. We only utilize the extracted RNA when both OD260/OD280 and OD260/OD230 values are greater than 1.8 and close to 2.0, ensuring the quality of the RNA used.

We justify the selection of hypoxanthine phosphoribosyl transferase (Hprt) as a reference gene due to its role as a housekeeping gene and its stable expression in RT-PCR assays using RNA samples extracted from tissues. We have employed Hprt as a reference gene in our prior publications (e.g., 'Mouse Model of Loeys-Dietz Syndrome Shows Elevated Susceptibility to Periodontitis via Alterations in Transforming Growth Factor-Beta Signaling. Satoru Yamada, Kenichiro Tsushima, Masaki Kinoshita, Hiromi Sakashita, Tetsuhiro Kajikawa, Chiharu Fujihara, Hang Yuan, Shigeki Suzuki, Takayuki Morisaki, Shinya Murakami. Frontiers in Physiology, Volume 12, Article 715687, 2021; 'Mice lacking PLAP-1/asporin counteract high-fat diet-induced metabolic disorder and alveolar bone loss by controlling adipose tissue expansion. Hiromi Sakashita, Satoru Yamada, Masaki Kinoshita, Tetsuhiro Kajikawa, Tomoaki Iwayama, Shinya Murakami. Scientific Reports, Volume 11, Issue 1, Article 4970, 2021).

Additionally, we have re-uploaded Table 1 for your reference.

  1. generation of the knockout mice needs to be described, even if briefly. To just refer to other reference on the procedure is not sufficient.

              Response: In response to the reviewer's comments, we have included a description of the PLAP-1 KO animal generation in the Materials and Methods section in our revised manuscript: " The genomic locus encoding the murine PLAP-1 gene was targeted by a replacement DNA construct, leading to disruptions in exon 2 and 3 caused by the neomycin resistance cassette. Two independent ES cell clones were utilized to produce chimeras capable of transmitting the targeted allele to the germline when mated with C57BL/6 females [15].”

  1. What software was used in statistics?

              Response: We used Statcel4 (OMS Publishing, Saitama, Japan). We have added that in the Materials and Methods section.

  1. Description of the inter-/ intra-examiner calibration needs to be addressed regarding the qualitative histological analysis. Furthermore, initials of the examiner needs to be written in this section. Was the examiner blinded?

              Response: Thank you for your valuable suggestions and concerns. In scientific experiments, it is of utmost importance that experimental data are consistently and accurately evaluated. We wholeheartedly pledge to uphold this standard. The quantitative analysis of tissue sections in this experiment is carried out by our evaluator, Dr. Masaki Kinoshita. Multiple individuals meticulously review the raw data in this experiment to ensure the precision of the quantitative experimental results. It is important to note that the evaluation process is not conducted in a blinded manner; however, Dr. Masaki Kinoshita conducts the quantitative analysis of tissue sections by referencing only the sample numbers, demonstrating a scrupulous commitment to prevent any arbitrary bias in the final experimental results.

We have incorporated the statement, “Each sample was measured repeatedly in three different sections of an individual by an examiner named MK. The average value was then calculated to represent the in-tensity of expression.” into the Materials and Methods section of the revised manuscript.

  1. The abstract needs to be divided into: background, materials and methods, results and conclusions. The authors need to include the statistical values as well in the abstract.

              Response: In accordance with the Reviewer’s comments, we have revised the abstract of the revised manuscript.

Comments on the Quality of English Language

Minor English language edits are needed, in terms of grammar.

              Response: We engaged a professional English proofreader before submitting our work. We submitted a Certificate of Editing, and we also meticulously proofread our revised manuscript.

Reviewer 2 Report

Comments and Suggestions for Authors

Manuscript “Mice lacking PLAP-1/Asporin show alteration of periodontal 2 ligament structures and acceleration of bone loss in periodontitis” aims to understand function of PLAP-1 in maintaining collagen fibrils in the PDL and explore its potential therapeutic application. Authors find that PLAP-1 knock out mice exhibit an enlarged PDL space with increased expression of Col3, BGN and DCN extracellular matrix proteins. Furthermore, highly severe alveolar bone resorption is observed in the ligature-induced periodontitis mouse model using PLAP-1 knockout mice. This manuscript will be interesting to the field of periodontitis. However, there are several questions need to be addressed.

Major

1.      In Figure 1E, authors try to show the increased PDL space in the PLAP-1 KO mice. However, the image in Figure 1E not clearly supports the conclusion. Can author provide coronal view of the CT and also provide a HE images to show the defects?

2.       In Figure 2A, it’s very hard to see the PDL in the sectioning. Could authors add some staining of PDL markers, so it will be clearly seen?

3.      In Figure 2, authors have analyzed the expression of Col1, Col3 and BGN. Authors conclude that expression of Col3, BGN and DCN increase in the PLAP-1 KO mice, but data of Col3 in figure 2 is not convincing. Could author provide more representative data and include an alternative approach for the data?

4.      In Figure 3A, authors measure the diameter of collagen fibrils using TEM. Can authors add scale bar to the images? Cells in the PLAP-1 KO mice seem larger than WT controls.

5.      In Figure 5, authors show that increased alveolar bone resorption is observed in PLAP-1 KO mice. However, expression of PLAP is not observed in alveolar bone (Figure 2C). Can authors discuss or explain the alveolar bone defects observed in the PLAP-1 KO mice?

Minor comments

1.      The magnifications of Figure 5D is not enough for the details. Can authors add images at higher magnification?

Author Response

Reviewer #2

Manuscript “Mice lacking PLAP-1/Asporin show alteration of periodontal 2 ligament structures and acceleration of bone loss in periodontitis” aims to understand function of PLAP-1 in maintaining collagen fibrils in the PDL and explore its potential therapeutic application. Authors find that PLAP-1 knock out mice exhibit an enlarged PDL space with increased expression of Col3, BGN and DCN extracellular matrix proteins. Furthermore, highly severe alveolar bone resorption is observed in the ligature-induced periodontitis mouse model using PLAP-1 knockout mice. This manuscript will be interesting to the field of periodontitis. However, there are several questions need to be addressed.

Major

  1. In Figure 1E, authors try to show the increased PDL space in the PLAP-1 KO mice. However, the image in Figure 1E not clearly supports the conclusion. Can author provide coronal view of the CT and also provide a HE images to show the defects?

Response: We appreciate your valuable suggestions and concerns. We have included new images providing a coronal view of the high resolution micro-CT scan and quantitative analysis of the periodontal ligament space area in the revised Figures 1G and 1H. HE images of PLAP-1 KO and WT mice have been incorporated to highlight the increased periodontal ligament space in the revised Figure 1I.

  1. In Figure 2A, it’s very hard to see the PDL in the sectioning. Could authors add some staining of PDL markers, so it will be clearly seen?

              Response: Thank you for your constructive suggestions and concerns. We have not conducted immunohistochemistry analysis of PDL markers, such as Periostin, in Figure 2A. To facilitate a clearer understanding of the PDL in Figure 2A, we have included an HE staining image of a WT animal in the revised Figure 2A.

  1. In Figure 2, authors have analyzed the expression of Col1, Col3 and BGN. Authors conclude that expression of Col3, BGN and DCN increase in the PLAP-1 KO mice, but data of Col3 in figure 2 is not convincing. Could author provide more representative data and include an alternative approach for the data?

              Response: We appreciate your valuable suggestions and concerns. In response to the reviewer's comments, we have replaced the immunohistochemistry images of Col3 to more clearly depict the differences in Col3 expression between WT and PLAP-1 KO mice in Figure 2C.

  1. In Figure 3A, authors measure the diameter of collagen fibrils using TEM. Can authors add scale bar to the images? Cells in the PLAP-1 KO mice seem larger than WT controls.

              Response: Thank you for the comment. We have added scale bars to Figure 3A in the revised manuscript.

  1. In Figure 5, authors show that increased alveolar bone resorption is observed in PLAP-1 KO mice. However, expression of PLAP is not observed in alveolar bone (Figure 2C). Can authors discuss or explain the alveolar bone defects observed in the PLAP-1 KO mice?

              Response: We appreciate your comment. In the Discussion section, on page 12, line 261, we have elaborated on this issue: "The PDL of PLAP-1 KO mice showed decreased strength and an enlarged PDL cavity, suggesting that occlusal trauma may have become more severe due to the increased alveolar bone resorption and an increased number of osteoclasts. PLAP-1 negatively regulates TRL2- and TRL4-mediated inflammatory responses in PDL cells [14]; PLAP-1 KO mice may lack this inhibitory mechanism, leading to the severity of periodontitis observed in this study. Further investigations are necessary to explore whether PLAP-1 KO mice develop periodontitis using occlusal trauma models and old-age mouse models.”

Minor comments

The magnifications of Figure 5D is not enough for the details. Can authors add images at higher magnification?

Response: We appreciate your valuable suggestions and concerns. We have obtained new higher magnification images that clearly depict TRAP-positive cells and have replaced the images in Figure 5D in the revised manuscript.

Round 2

Reviewer 1 Report

Comments and Suggestions for Authors

The authors present a revised version of their mansucript. The manuscript has significantly improved from its previous version, and the authors have successfully addressed the comments and questions raised by the reviewer. As such, this manuscript can be recommended for publication in its current form in the International Journal of Molecular Sciences. 

Reviewer 2 Report

Comments and Suggestions for Authors

No further comments